# Tolerability of Oral Supplementation with Microencapsulated Ferric Saccharate Compared to Ferrous Sulphate in Healthy Premenopausal Woman: A Crossover, Randomized, Double-Blind Clinical Trial

**DOI:** 10.3390/ijms232012282

**Published:** 2022-10-14

**Authors:** Marina Friling, Ana María García-Muñoz, Tania Perrinjaquet-Moccetti, Desirée Victoria-Montesinos, Silvia Pérez-Piñero, María Salud Abellán-Ruiz, Antonio J. Luque-Rubia, Ana Isabel García-Guillén, Fernando Cánovas, Eran Ivanir

**Affiliations:** 1IFF Health, Migdal Haemeq 23106, Israel; 2Health Science Department, Campus de los Jerónimos, Universidad Católica San Antonio de Murcia UCAM, 30107 Murcia, Spain; 3IFF Health, 8820 Wädenswil, Switzerland

**Keywords:** iron supplementation, AB-Fortis^®^, microencapsulated iron, ferric saccharate, ferrous sulphate, gastrointestinal side effects, tolerability

## Abstract

A single-center, crossover, randomized, double-blind, and controlled clinical study was conducted to assess the tolerability profile, especially with regard to gastrointestinal complaints, of oral supplementation with AB-Fortis^®^, a microencapsulated ferric saccharate (MFS), as compared with conventional ferrous sulphate (FS) in healthy premenopausal women. A dose of 60 mg/day of elemental iron was used. The test products were administered for 14 consecutive days with a washout period of two menstrual episodes and a minimum of one month between the two intervention periods. The subjects completed simple-to-answer questionnaires daily for 14 days during both the intervention and the washout periods, capturing the symptoms associated with oral iron supplementation and overall health aspects. Following product consumption, the incidences of symptoms, numbers of complaints/symptoms, overall intensity, and total days with symptoms were found to be significantly higher for FS consumption as compared to MFS. The better tolerability profile of MFS over FS was further substantiated when both products were compared to a real-life setting (i.e., the washout period). Overall, the administration of both study products was safe with no serious or significant adverse events reported. In summary, the current study shows the better tolerability of the MFS preparation when compared to that of the FS, presenting MFS as a well-tolerated and safe option for improving iron nutrition.

## 1. Introduction

Iron is an essential element for living organisms. It is crucial for multiple metabolic processes, including oxygen transport, DNA synthesis, oxygenation of muscles, and various oxidation–reduction reactions [1,2]. Excluding specific diseases, iron status represents a balance between iron input and physiological loss [3,4]. Food is the only natural source of iron, and the amount consumed should replace iron daily losses through the intestine, genitourinary tract, and skin [3]. Maintaining iron balance is most important under increased iron requirements due to pregnancy, menstruation, etc. [3,5].

Dietary iron is present in varying concentrations in a broad range of foods; however, the bioavailability of iron differs depending on the food sources and the types of dietary iron [6,7]. Heme iron is highly bioavailable, and it is found in meat and other animal products. Non-heme iron is derived from plant sources (mainly cereals, legumes, and vegetables); its absorption is much lower, and it is strongly influenced by the presence of other food components, such as polyphenols, tannins, and phytate [3,8,9]. Consequently, iron intake and heme/non-heme iron proportions are crucial to avoid iron deficiency anemia. According to the World Health Organization (WHO), 24.8% of the world population is anemic, and iron deficiency is estimated to be a responsible factor for approximately 50% of anemia cases [10,11,12].

Iron-fortified foods and supplements with iron alone or in combination with folic acid and/or other vitamins and minerals have been an extensive focus of research, particularly in the management of iron deficiency among vulnerable groups (low-income populations, children, and pregnant and postpartum women) [13,14,15,16]. Although iron supplements are widely available, the major difficulty encountered with oral iron intake is gastrointestinal (GI) tolerability, mainly constipation or diarrhea, nausea/vomiting, and epigastric distress [17,18]. GI side effects are recognized as one of the main factors associated with low adherence to iron supplementation, and adjusted oral regimens (e.g., weekly or three times a week instead of daily administration) have been proposed to increase GI tolerability and to improve compliance [19,20,21,22,23]. Results from iron supplementation studies suggest that a greater number of adverse events (AEs) were associated with the immediate release of ferrous sulphate (FS), ferrous fumarate, or ferrous gluconate formulations [17,18]. Although it is generally considered that doses of 50–60 mg iron/day generate fewer side effects than higher doses, no evidence has been found for the dose-response effect or threshold, whether considered as an amount of iron per day or per dose [17].

Microencapsulation of iron has been suggested as being a better alternative compared to conventional iron. The benefits may include the prevention of unpleasant organoleptic characteristics and GI side effects and the alteration of the interaction of iron with the food matrix to overcome intraluminal absorption inhibitors and improve the bioavailability of non-heme iron [18,24,25,26]. Given the promising benefits of microencapsulated iron in reducing AEs, the present clinical trial was designed to assess the tolerability, in particular the GI side effects, of a daily dose of 60 mg of AB-Fortis^®^, a microencapsulated iron formulation (ferric saccharate, MFS), as compared to the same dose of a conventional ferrous sulphate iron supplement.

## 2. Results

### 2.1. Study Population and Compliance

A total of 68 women were assessed for eligibility. Fifty-one (75%) met the inclusion and exclusion criteria and were randomized to supplementation with MFS (n = 26) and FS (n = 25). Two participants assigned to the FS group dropped out early due to COVID-19 and were excluded from both the tolerability and the safety analysis. Another woman assigned to the MFS group abandoned the study during the washout period, and a further one was excluded because of poor compliance during the consumption period of both products (14.5% capsules intake). Both participants were excluded from the analysis according to the predefined protocol violation. As shown in Figure 1, the safety analysis and tolerability analysis included 49 and 47 participants, respectively. Table 1 lists the baseline characteristics of the participants included in the tolerability analysis. The participants were Caucasian, with a mean age of 30.7 ± 7.4 years and Body Mass Index (BMI) of 22.3 ± 1.8 kg/m^2^. The baseline iron profile included mean hemoglobin level of 13.3 ± 0.8 g/dL, ferritin of 30.5 ± 19.6 µg/L, transferrin saturation of 28.6 ± 11.9%, and serum iron of 93.0 ± 36.6 µg/dL. Ten participants were current smokers, and 19 showed a serum ferritin level > 30 µg/L. The two sequences of random allocation were matched for all the sociodemographic and baseline characteristics, besides hemoglobin levels that were found to be statistically significantly different (*p* < 0.05); however, the difference was not considered to be clinically relevant. The compliance was high for both the MFS (97.5 ± 4.5%) and the FS (97.9 ± 4.4%) products, based on the evaluation of the unused capsules.

### 2.2. Primary Outcome

The percentage of participants who experienced at least one symptom was 72.3% during the consumption of MFS, 91.5% during the consumption of FS, and 74.5% during the washout period, with significant differences in favor of MFS consumption, which showed lower percentages as compared to those of FS (*p* = 0.012). The percentage of participants reporting at least one symptom was significantly higher during the consumption of FS as compared with the washout period (*p* = 0.039) as well. No significant difference was found in the proportion of subjects reporting adverse events between MFS and the washout period (Table 2). 

### 2.3. Secondary Outcomes

#### 2.3.1. Frequency of Symptoms

The percentage of participants reporting symptoms classified as gastrointestinal was significantly lower in subjects receiving the MFS supplement as compared to the FS (68.1% vs. 87.2%; *p* = 0.012). The frequencies of nausea, abdominal pain, flatulence/abdominal swelling, and diarrhea were significantly lower in favor of MFS vs. FS. The occurrence of overall GI-related symptoms, as well as nausea, heartburn, flatulence/abdominal swelling, and diarrhea, was significantly higher during the FS supplementation than during the washout period (Table 3). 

#### 2.3.2. Number of Complaints/Symptoms

During the 14-day MFS supplementation period, the participants recorded a significantly lower number of complaints and symptoms related to iron consumption compared to those of FS. The number of complaints and symptoms was also significantly higher for FS vs. the washout period. The significant findings were for both the total and the GI-related complaints/symptoms (Table 4).

#### 2.3.3. Intensity of Symptoms

The results of the overall daily intensity of the symptoms are presented in Table 5. Significantly lower intensity scores were observed for nausea, abdominal pain, and flatulence/abdominal swelling for MFS compared with FS. The overall daily intensity of nausea and flatulence/abdominal swelling was significantly higher during the supplementation with FS as compared with the washout period. Moreover, significantly higher overall daily intensity of heartburn was recorded for MFS as compared to the washout period. However, this difference was considered not clinically meaningful due to the higher intensity level observed with FS, which was found to be not significantly different from the washout period. Regarding acute intensity of symptoms, there were no significant differences between the MFS and the FS supplements, and only flatulence/abdominal swelling showed a significantly higher visual analogue scale (VAS) score for both MFS and FS as compared with the washout period. 

#### 2.3.4. Duration of Symptoms

The results of the overall duration of symptoms (days) are presented in Table 6. Significant differences between MFS and FS were found for nausea, abdominal pain, flatulence/abdominal swelling, and diarrhea, with a lower number of days for the MFS supplement. The total duration of symptoms was significantly higher for nausea, flatulence/abdominal swelling, diarrhea, and metallic taste following FS consumption as compared to the washout period. The results of the mean daily duration of symptoms (minutes) showed significant differences only for headache in the comparison between the MFS and the FS supplementation (75.9 ± 10.5 vs. 251.9 ± 412.2 min; *p* = 0.015) and for heartburn in the comparison between the FS supplementation and the washout period (127.5 ± 165.4 vs. 28.5 ± 59.4 min; *p* = 0.036). No other significant differences were found between study periods in terms of daily symptom duration.

#### 2.3.5. Health Status, Daily Activity, Bowel Movement, Stool Consistency

No significant differences were observed in health status, impact on daily activities, or in the bowel movements, with no abnormal events reported during the consumption days of both products and the washout period. The stool consistency values were all in the normal range, between a type 3 (like a sausage but with cracks on its surface) and 4 (like a sausage or a snake, smooth and soft) of the Bristol stool scale. The mean values were 3.4 ± 0.9, 3.6 ± 0.9, and 3.3 ± 0.7 for the MFS, FS, and washout period, respectively. A significant difference was found only between the washout period and the FS group, although the observed changes were relatively small.

#### 2.3.6. Sleep Duration and Quality

The sleep duration values were all normal, with the participants reporting good sleep quality. Sleep duration was reported as 7.2 ± 0.9 h during MFS and FS consumption and 7.1 ± 0.8 h during the washout period. The mean values for sleep quality were 2.0 ± 0.4, 2.2 ± 0.5, and 2.1 ± 0.4 in terms of a 5-point Likert scale for MFS, FS, and the washout period, respectively. Although minor changes were recorded with sleep parameters, statistically significant differences were observed in sleep duration between MFS and the washout period and in sleep quality when comparing between MFS and both FS and the washout period.

#### 2.3.7. Iron Parameters

In relation to iron parameters, both the MFS and the FS groups remained in the normal range. A statistically significant difference was observed in the mean ferritin value at the end of the phase following the FS consumption as compared to the MFS (40.17 ± 23.07 vs. 34.11 ± 24.87; *p* = 0.024). There were no statistically significant differences between the study products (FS vs. MFS) at the end-of-phase values in hemoglobin (13.57 ± 0.64 vs. 13.63 ± 0.69), transferrin saturation (27.44 ± 14.57 vs. 27.22 ± 10.98), and serum iron levels (86.56 ± 38.85 vs. 89.03 ± 29.06). 

### 2.4. Safety Evaluation

None of the participants discontinued the study due to AEs, and no serious AE was reported. Sixteen participants (32.7%) experienced at least one AE at each of the consumption periods, MFS or FS, which were of mild severity in all cases. During the washout period, the percentage of participants with at least one AE was 18.4% (n = 9). A total of 14 participants (28.6%) reported an AE possibly related to the study product, 7 (14.3%) during supplementation with MFS, and 11 (22.4%) during supplementation with FS. Dark stools was the most common AE in all the iron supplement groups. No clinically significant changes in laboratory values, anthropometrics, physical examination, or vital signs over time were observed.

## 3. Discussion

Iron supplementation or fortification is often the chosen strategy in the management of iron deficiency and iron-deficiency anemia [27,28,29,30]. Among the different types of oral iron preparations, FS is the most popular form but carries a major risk of noncompliance due to side effects, which are mostly associated with gastrointestinal disturbances [17,18,21,22,23]. The suggestion that the microencapsulation of iron provides a better protection and improves overall tolerability compared to conventional iron highlights the need to search for alternative solutions. The present findings confirmed the hypothesis of the study that the consumption of an iron supplement in the formulation of a microencapsulated ferric preparation was associated with a better tolerability profile as compared to ferrous sulphate, a commonly used iron supplement. Better tolerability of the MFS product was observed in the primary outcome and in different variables of the secondary outcomes. During the periods of consumption of the MFS supplement, a lower percentage of participants experienced clinical manifestations and overall GI events as well as individual symptoms of nausea, abdominal pain, flatulence/abdominal swelling, and diarrhea, with statistically significant differences as compared with the periods of FS consumption. The number of complaints/symptoms reported, the overall intensity, and the total days with symptoms were also more favorable for the MFS supplement. The better tolerability profile of MFS over FS was further substantiated when both products were compared to the washout period. The comparative analysis showed that the tolerability profile of MFS consumption was comparable to a real-life setting (i.e., without iron supplementation) as opposed to that of FS. Among other study variables, the consumption of MFS does not seem to have any impact on health status, daily activity, bowel movement, and stool consistency as compared to FS. Although, significant improvements in sleep parameters were observed, these differences were small and not clinically relevant. In relation to iron profile, both the MFS and the FS groups remained in the normal range, with a small increase in ferritin value following the FS supplementation. Overall, the administration of both study products was safe, with reported AEs being of mild intensity and without clinically significant changes in laboratory values, anthropometrics, physical examination, or vital signs. 

Commonly, oral iron supplements are digested and absorbed by the gastrointestinal tract, and the main sites of absorption are the small intestines. Iron in the ferrous form (Fe^2+^) is directly absorbed in the duodenum by a divalent metal transporter (DMT-1), while ferric iron (Fe^3+^) is reduced to the ferrous form by a ferric reductase at the apical surface of the epithelial brush border [2,31,32]. Excess oral iron in the intestinal tract usually induces peroxidative damage through production of reactive oxygen species. As a result, the intestinal epithelial cells are damaged, leading to incompleteness of the intestinal mechanical barrier. This damage may cause inflammation of the mucosa which in turn decreases iron absorption by stimulating hepcidin production and secretion and decreasing DMT-1 protein levels. Furthermore, excess iron enhances the growth of pathogenic intestinal microbiota species, resulting in a microbial imbalance that may trigger a series of side effects, such as diarrhea, abdominal pain, vomiting, etc. [23,33]. In this regard, the MFS formulation keeps excess iron from interacting with the intraluminal environment and increases the compatibility with other food ingredients, thus reducing the typical iron-related adverse effects, especially those of GI complaints. The results of the present study are aligned with other reported studies, showing lower incidences of GI side effects with controlled-release iron formulations as compared with the ferrous salt preparations [18].

The bioavailability of iron formulation is a major requirement when designing alternatives for common iron supplements such as FS. Previous studies on the MFS formulation have shown the bioavailability of this compound to be similar to that of the FS preparation, in both experimental and human studies, which supports the background evidence for the design of the present trial. In an experiment with male Sprague–Dawley rats, the animals with iron deficiency anemia were treated with an iron-deficient diet for 14 days with or without the addition of iron in the form of FS or MFS [34]. No differences were observed between the two fortified groups in body weight, feed efficiency, mean corpuscular volume of reticulocytes, and average hemoglobin content in the reticulocytes, with significant differences compared to the negative control group. Overall, this study showed that MFS displayed a similar bioavailability to that of FS at the dose tested; so, the ingestion of MFS is as effective as that of FS in the recovery from iron deficiency anemia in this model [34]. Similar findings were observed in a crossover, randomized, double-blind study, comparing the absorption of iron from MFS and FS in a fortified milk product in 17 healthy subjects [35]. No significant differences in serum iron and transferrin saturation were found between the investigational products, suggesting that the iron absorption of MFS was similar to that of FS [35].

A daily dose of 60 mg of elemental iron was used, which was considered to be effective in fulfilling the anticipated goals of the study. The dose of 60 mg/day of elemental iron is included in the 2016 WHO guideline recommendation for adult women and adolescent girls [36]. Moreover, we used a short and simple-to-answer symptom questionnaire that was based on Pereira et al. and was designed to assess GI symptoms after oral FS supplementation [37]. The questionnaire was completed by the participants during the 14 days of the intervention period and the washout interval, the aim of which was to capture the occurrence and intensity of the predefined symptoms commonly related to the use of oral iron preparations, particularly FS. The crossover comparator-controlled randomized design was considered appropriate for the evaluation of subjective complaints associated with interventions, in which participants serve as their own control.

The present findings, however, should be interpreted taking into account the limitations of the study. The main limitation is related to the selection criteria, according to which premenopausal women aged 18–50 years were selected, so the study findings are applicable to women with these characteristics. It should be noted that the study population consisted of Caucasian women, which may be explained by the setting in which participants were recruited (Murcia Region, Spain) and the number of participants enrolled. It would be of interest to assess the reproducibility of the study findings in larger samples of participants from other ethnic groups. In relation to the control product, ferrous sulphate rather than a matched substance correspondent (iron saccharate) was selected because it is the most commonly used active ingredient for iron supplementation. However, despite the strict inclusion criteria and the limitations of the external validity of the data obtained in the randomized controlled trials, the study findings are clinically relevant and have direct applicability for the use of MFS as a nutritional supplement to restore iron levels.

## 4. Materials and Methods

### 4.1. Study Design

This was a double-blind, 2-way crossover, randomized (ratio 1:1), comparator-controlled clinical trial conducted in healthy, volunteer, premenopausal women at the Health Sciences Department of Universidad Católica San Antonio de Murcia (UCAM), Murcia, Spain, between 1 February and 23 December 2020. Each subject participated in two 14-day intervention periods separated by a washout period of two consecutive menstrual episodes, with a minimum of one month between the two intervention periods (Figure 2). The study was conducted in accordance with the Declaration of Helsinki and Good Clinical Practice (GCP) standards [38,39] and was registered at ClinicalTrials.gov (NCT04199234). All the participants gave their written, informed consent and were told that they could withdraw from the study at any time. 

### 4.2. Participants

The participants were mainly recruited by advertising the study through the UCAM community email list. Eligible participants were premenopausal women aged between 18 and 50 years with regular menstrual cycles (28 ± 5 days) and normal health as established by medical history, vital signs, physical examination, electrocardiogram (ECG), and laboratory data within normal limits, including complete blood cell count, biochemical profile, and urinalysis. Other inclusion criteria were BMI between 20 and 25 kg/m^2^, non-anemic, C-reactive protein (CRP) < 5 mg/L, and willingness to maintain stable nutritional and general habits during the study. Exclusion criteria were serum hemoglobin levels < 12 g/dL, hyperlipidemia (low-density lipoprotein (LDL) > 130 mg/dL and/or triglycerides > 200 mg/dL), severe premenstrual symptoms, relevant comorbid diseases (e.g., peptic ulcer, ulcerative colitis or enteritis, inflammatory bowel disease, iron storage disorders, liver and renal dysfunction, etc.), chronic conditions, infectious disease or any active medical illness in the 48 h prior to the baseline visit, drug or alcohol abuse, history of gastric bypass or bariatric surgery, use of nutritional supplements within the previous month, intake of iron supplements and/or iron-containing multivitamins during the three months prior to study entry, use of chronic medication (except oral contraceptives), previous participation in iron tolerability trials, participation in a clinical research trial within 30 days of randomization, pregnancy or breastfeeding, planning pregnancy, intolerance to iron supplements, following a specific diet 30 days prior to the start of the study (e.g., high protein diet), blood donation in the previous month, cognitive impairment or incapability to give informed consent, and any other laboratory abnormality, medical condition, or psychiatric disorder that, in the opinion of the investigator, may adversely affect the ability of the subject to complete the study or its measurements or that may pose a significant risk to the subject.

### 4.3. Study Protocol and Data Collection

At the screening visit (V0), the subjects were initially screened and given the informed consent form. Once consent was obtained, demographic information and smoking status were collected, the inclusion and exclusion criteria were reviewed, including medical history, physical examination, anthropometric measurements (weight, height), heart rate, and blood pressure, ECG, blood tests, urinalysis, and a pregnancy test. The blood tests included a complete blood count with differential, hemoglobin, hemoglobin A1c (HbA1c), glucose, electrolytes (sodium, potassium, chloride), liver function tests (alanine aminotransferases (ALT) and aspartate aminotransferases (AST)), bilirubin, gamma-glutamyl transferase (GGT), creatinine, blood urea nitrogen, glomerular filtration rate, uric acid, albumin, total proteins, CRP, lipid profile (total cholesterol, triglycerides, high-density lipoprotein (HDL), and LDL), ferritin, transferrin saturation, and serum iron. Anthropometrics, vital signs, physical examination, AEs, blood (after 12 h fasting), and urine samples were taken at each of the study visits (V0-V4), with the same variables recorded. At clinic visit 1 (V1), the subjects were randomized and received the exact number of capsules required for the 14-day intervention period (MFS or FS). The subjects were given a diary to record consumption time and complete daily questionnaires during the 14 consumption days.

The daily questionnaires collected information on the previous 24 h for the following variables: (a) daily occurrence of nausea, vomiting, heartburn, abdominal pain, flatulence/abdominal swelling, diarrhea, constipation, metallic taste, headache, and shortness of breath, all of which were categorized as yes (presence) or no (absence); (b) overall daily intensity of nausea, heartburn, abdominal pain, flatulence/abdominal swelling, using a 0–10 cm VAS (no upset to severe) or number of occurrences for diarrhea and vomiting; (c) acute intensity (at the time of recording) of nausea, heartburn, abdominal pain, flatulence/abdominal swelling, vomiting, diarrhea, constipation, metallic taste, headache, and shortness of breath using a 0–10 cm VAS (no upset to severe); (d) daily duration of nausea, heartburn, abdominal pain, flatulence/abdominal swelling, metallic taste, headache, and shortness of breath, measured in minutes; (e) daily health status using a 0–10 cm VAS (very bad to very good); (f) daily health status with reference to the previous day using a 0–10 cm VAS (much worse to much better); (g) impact of symptoms on daily activities, categorized as yes (impact) or no (no impact); (h) sleep quality using a 5-point Likert scale (very relaxing to trouble falling and/or staying asleep); (i) sleep duration of the previous night measured in hours and minutes; (k) stool consistency using the 7-point Bristol stool form scale [40]. The subjects returned to the clinic (visit 2; V2) following 14 days of product consumption (Day 15). Both visits were similar in assessments and procedure. 

Compliance with the study products was assessed by counting the empty and non-empty blister packages returned. A subject was considered compliant when ≥ 80% of the capsules provided had been consumed. AEs were recorded after the intervention periods by questioning and direct participant reporting. There was a minimum of a washout period of two consecutive menstrual episodes, with a minimum of a one-month period before the crossover product was given and the above procedure was repeated (V3–V4). Throughout the washout period, the participants continued recording the daily questionnaires based on the previous 24 h.

### 4.4. Study Products

The investigational product MFS, trademarked as AB-Fortis^®^ (IFF Health, Londerzeel, Belgium) is a patented microencapsulation system consisting of a calcium alginate matrix that holds and protects a salt of ferric saccharate inside. MFS is produced by ionotropic gelation of alginate with calcium, entrapping the iron salt inside. Calcium displays a stronger interaction with alginate, stabilizing the microcapsules and avoiding the release of iron. The interaction of iron with alginate is weaker and so it is released when the calcium layer is dissolved in the intestine [35]. Both the MFS and the FS powders were encapsulated by an independent company (Laboratorios Admira, Murcia, Spain), providing an identical capsule format, containing 60 mg of elemental iron. The participants were instructed to take one capsule a day of the study product, 2 h before lunch for the two 14-day periods.

### 4.5. Randomization and Blinding

Randomization was carried out with computer-generated randomization lists. The allocation was stratified by serum ferritin levels (≤ or >30 µg/L) and current smoking habits (smokers or non-smokers), using the Epidat 3.1 program (Xunta de Galicia, Santiago de Compostela, Spain). The study products were labeled with an individual randomization number to ensure that the study was double-blind. The identity of the specific product was blinded to the participants, study staff, and investigators. Copies of the individual sealed envelopes containing the unblinded codes for each participant were available to the principal investigator. The investigators and study staff remained blinded to the study group assignments until the database was locked.

### 4.6. Outcomes

The primary outcome was the proportion of participants who experienced at least one AE during the period of consumption of MFS and FS, considering the following symptoms: nausea, heartburn, abdominal pain, flatulence/abdominal swelling, diarrhea, vomiting, metallic taste, constipation, shortness of breath, and headache. The secondary outcomes included the prevalence of participants who reported a specific clinical symptom, total number of symptoms/incidences, overall daily intensity of symptoms and acute intensity of symptoms, daily duration and total duration of symptoms, daily health status, impact of symptoms on daily activities, number of bowel movements, stool consistency, sleep quality/duration, and iron profile parameters. The safety endpoints were changes in the vital signs, anthropometrics, laboratory parameters, physical examination, and occurrence of AEs not defined under the tolerability diary measurement.

### 4.7. Statistical Analysis

A sample size of 45 participants per group (MFS and FS), with a total of 48 assuming a 5% loss to follow-up, was estimated to detect a difference of 0.18 in the proportion of symptoms, with 80% statistical power and a two-sided significance level of 5%, based on a McNemar’s test of equality or paired proportions. SPSS version 21.0 (IBM Corp., Armonk, NY, USA) was used for statistical evaluation. The categorical data are expressed as frequencies and percentages and continuous variables as mean and standard deviation (SD). The comparisons between different periods used McNemar’s testing for categorical variables and a paired t-test for continuous variables. For variables with non-normal distribution according to the Kolmogorov–Smirnov test, the Wilcoxon signed-rank was used. Probability values of *p* < 0.05 were considered statistically significant. As part of the tolerability evaluation, the washout period was considered as a reference to the subject’s sensations in real life (without iron supplementation) and was compared to the two iron supplementation periods. Out of the washout period, a time window of 14 days was taken into consideration for the comparative analysis, matching the time frame of the MFS and FS consumption periods. The selected time window started from the 10th day into the washout period, which was considered a sufficient washout period for iron supplementation [37]. The analyzed subset was defined during the blind data review before the database was locked. The participants who attended the end-of-study visit and consumed at least 80% of the assigned study product were included in the tolerability analysis. All safety endpoints were analyzed based on the participants for whom any post-randomization safety data were available.

## 5. Conclusions

The present study shows that the administration of a nutritional supplement of MFS to healthy premenopausal women was associated with a better tolerability profile, especially in relation to GI side effects, as compared with a standard FS-based supplementation. These findings are clinically relevant in daily practice for improving iron supplementation.

## Figures and Tables

**Figure 1 ijms-23-12282-f001:**
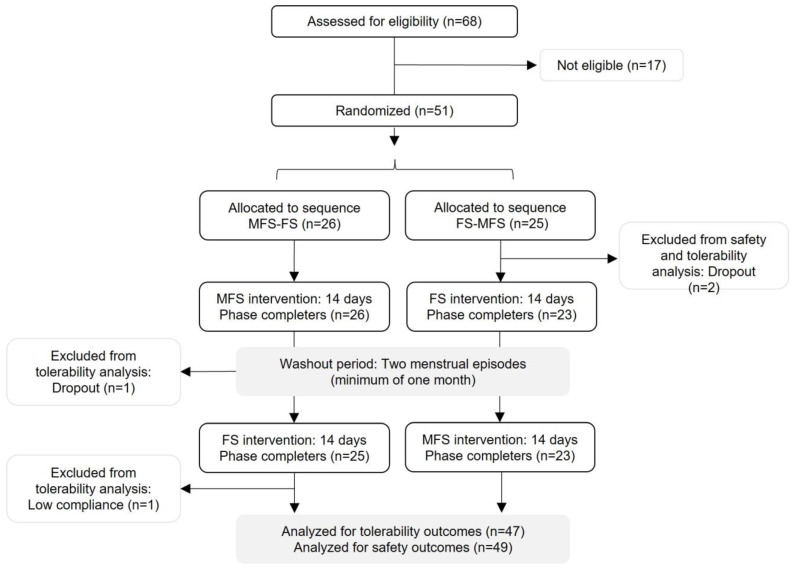
Flow chart of the study population (MFS: microencapsulated ferric saccharate, FS: ferrous sulphate).

**Figure 2 ijms-23-12282-f002:**
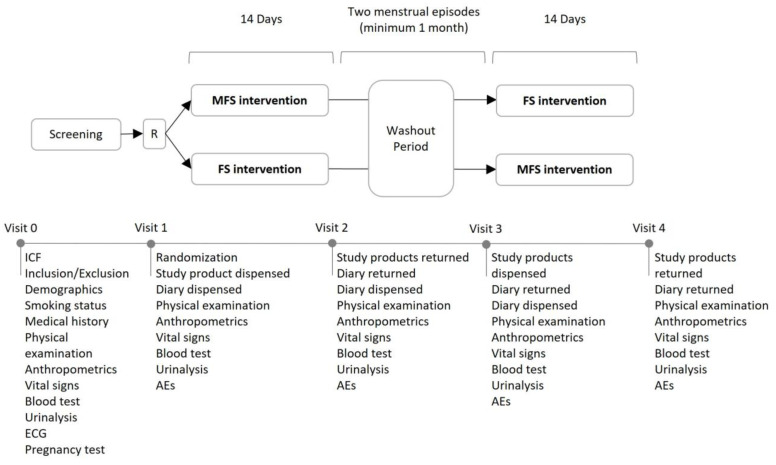
Overview of study design. This was a crossover, randomized, double-blind study. The two intervention phases were separated by washout period. The diary included daily questionnaires and product consumption time. (MFS: microencapsulated ferric saccharate, FS: ferrous sulphate, R: randomization, AEs: adverse events, ICF: informed consent form, ECG: electrocardiogram).

**Table 1 ijms-23-12282-t001:** Demographics and baseline characteristics of participants.

Characteristics	Total (n = 47)	Sequence	
		MFS-FS (n = 24)	FS-MFS (n = 23)
Age (years)	30.7 (7.4)	30.6 (6.9)	30.8 (8.1)
Ethnicity, n (%)			
Caucasian	47 (100.0)	24 (100.0)	23 (100.0)
BMI (kg/m^2^)	22.3 (1.8)	22.3 (1.7)	22.4 (1.9)
Heart rate (beats/min)	73.0 (12.9)	72.1 (14.6)	74.0 (11.2)
SBP (mmHg)	114.5 (12.0)	111.5 (10.3)	117.5 (13.2)
DBP (mmHg)	70.5 (7.6)	70.8 (7.3)	70.3 (8.1)
Current smokers, n (%)	10 (21.3)	6 (25.0)	4 (17.4)
Ferritin level > 30 µg/L, n (%)	19 (40.4)	9 (37.5)	10 (43.5)
Hemoglobin (g/dL)	13.3 (0.8)	13.1 (0.7)	13.5 (0.7)
Ferritin level (µg/L)	30.5 (19.6)	27.8 (15.6)	33.3 (23.1)
Transferrin saturation (%)	28.6 (11.9)	27.2 (12.8)	30.0 (11.0)
Serum iron (µg/dL)	93.0 (36.6)	86.3 (38.1)	100.0 (95.1)

Data expressed as mean (standard deviation) unless otherwise stated; SBP: systolic blood pressure, DBP: diastolic blood pressure; BMI: body mass index.

**Table 2 ijms-23-12282-t002:** Number and percentage of participants experiencing at least one symptom related to iron consumption.

Participants	Study Period			*p* Values *		
	MFS	FS	Washout	P1	P2	P3
Symptoms experienced	34 (72.3)	43 (91.5)	35 (74.5)	0.012	1.000	0.039
No symptoms experienced	13 (27.7)	4 (8.5)	12 (25.5)			

Data expressed as number of participants (%); evaluated symptoms included nausea, heartburn, abdominal pain, flatulence/abdominal swelling, diarrhea, metallic taste, constipation, vomiting, headache, and shortness of breath. * *p* value calculated based on McNemar’s test. *P1* = comparison between MFS and FS; *P2* = comparison between MFS and washout; *P3* = comparison between FS and washout; significant differences at *p* < 0.05.

**Table 3 ijms-23-12282-t003:** Number and percentages of participants who experienced GI-related symptoms, headache, and shortness of breath.

Symptoms	Study Period			*p* Values *		
	MFS (n = 47)	FS (n = 47)	Washout (n = 47)	P1	P2	P3
GI-related symptoms	32 (68.1)	41 (87.2)	30 (63.8)	0.012	0.774	0.003
Nausea	4 (8.5)	11 (23.4)	2 (4.3)	0.016	0.687	0.022
Heartburn	7 (14.9)	10 (21.3)	3 (6.4)	0.508	0.219	0.039
Abdominal pain	10 (21.3)	17 (36.2)	15 (31.9)	0.039	0.302	0.815
Flatulence/swelling	20 (42.6)	30 (63.8)	19 (40.4)	0.013	1.000	0.007
Diarrhea	6 (12.8)	14 (29.8)	5 (10.6)	0.039	1.000	0.012
Metallic taste	3 (6.4)	6 (12.8)	1 (2.1)	0.375	0.500	0.063
Constipation	11 (23.4)	12 (25.5)	9 (19.1)	1.000	0.804	0.629
Vomiting	0 (0.0)	1 (2.1)	0 (0.0)	1.000	NA	1.000
Headache	17 (36.2)	24 (51.1)	20 (42.6)	0.143	0.664	0.523
Shortness of breath	1 (2.1)	2 (4.3)	0 (0.0)	1.000	1.000	0.500

Data expressed as number of participants (%); NA: not applicable. GI-related symptoms included nausea, heartburn, abdominal pain, flatulence/abdominal swelling, diarrhea, metallic taste, constipation, and vomiting. * *p* value calculated based on McNemar’s test. *P1* = comparison between MFS and FS; *P2* = comparison between MFS and washout; *P3* = comparison between FS and washout; significant differences at *p* < 0.05.

**Table 4 ijms-23-12282-t004:** Number of complaints and symptoms related to iron consumption.

Reports	Subjects ^†^	Study Period			*p* Values *		
		MFS	FS	Washout	P1	P2	P3
Number of complaints							
Total	45	4.6 (5.1)	8.8 (8.1)	3.5 (3.7)	<0.001	0.169	<0.001
GI-related	42	4.0 (4.3)	7.7 (6.7)	2.8 (3.4)	0.001	0.105	<0.001
Number of symptoms							
Total	45	1.8 (1.5)	2.8 (1.7)	1.6 (1.3)	<0.001	0.646	<0.001
GI-related	42	1.5 (1.2)	2.4 (1.4)	1.3 (1.1)	<0.001	0.426	<0.001

Data expressed as mean (standard deviation); Total complaints/symptoms included nausea, heartburn, abdominal pain, flatulence/abdominal swelling, diarrhea, metallic taste, constipation, vomiting, headache, and shortness of breath. GI-related complaints/symptoms included all complaints/symptoms considered in the total category, excluding headache and shortness of breath. ^†^ Subjects without any related reports were excluded from the analysis. * *p* values are based on paired *t*-test. *P1* = comparison between MFS and FS; *P2* = comparison between MFS and washout; *P3* = comparison between FS and washout; significant differences at *p* < 0.05.

**Table 5 ijms-23-12282-t005:** Overall daily intensity of symptoms related to iron consumption.

Symptoms	Subjects ^†^	Study Period			*p* Values *		
		MFS	FS	Washout	*P1*	*P2*	*P3*
Nausea (cm)	13	0.8 (1.6)	3.4 (2.6)	0.6 (1.5)	0.006	0.752	0.039
Heartburn (cm)	13	1.2 (1.4)	2.1 (2.5)	0.5 (1.1)	0.507	0.017	0.062
Abdominal pain (cm)	26	1.6 (2.5)	3.0 (2.9)	2.7 (2.6)	0.010	0.179	0.677
Flatulence/swelling (cm)	34	2.2 (2.5)	3.1 (2.6)	2.1 (2.5)	<0.002	0.785	0.037
Diarrhea (events/day)	17	1.0 (1.6)	2.0 (1.2)	1.1 (1.9)	0.122	1.000	0.052
Vomiting (events/day)	1	0.0 (0.0)	2.5 (0.0)	0.0 (0.0)	NA	NA	NA

Data expressed as mean (standard deviation); NA: not applicable. ^†^ Subjects without any related reports were excluded from the analysis. * *p* values are based on the Wilcoxon signed rank test. *P1* = comparison between MFS and FS; *P2* = comparison between MFS and washout; *P3* = comparison between FS and washout; significant differences at *p* < 0.05.

**Table 6 ijms-23-12282-t006:** Overall duration of symptoms related to iron consumption.

Symptoms	Subjects ^†^	Study Period			*p* Values *		
		MFS (days)	FS (days)	Washout (days)	*P1*	*P2*	*P3*
Nausea	13	0.3 (0.5)	2.2 (2.2)	0.1 (0.4)	0.005	0.414	0.008
Heartburn	13	1.4 (1.7)	1.5 (1.4)	0.5 (1.1)	0.843	0.065	0.051
Abdominal pain	26	0.8 (1.2)	1.7 (2.2)	1.0 (1.1)	0.017	0.552	0.208
Flatulence/swelling	34	2.3 (3.4)	3.9 (3.6)	1.6 (2.7)	0.008	0.174	<0.001
Diarrhea	17	0.7 (1.2)	1.6 (1.3)	0.4 (0.8)	0.025	0.160	0.003
Metallic taste	7	0.4 (0.5)	4.0 (4.1)	0.1 (0.4)	0.058	0.157	0.042
Constipation	25	1.3 (1.8)	1.5 (2.0)	1.0 (1.7)	0.924	0.569	0.419
Vomiting	1	0.0 (0.0)	2.0 (0.0)	0.0 (0.0)	NA	NA	NA
Headache	36	1.1 (1.7)	1.9 (2.9)	1.1 (1.3)	0.101	0.876	0.198
Shortness of breath	3	0.3 (0.6)	0.7 (0.6)	0.0 (0.0)	0.564	0.317	0.147

Data expressed as mean (standard deviation); NA: not applicable. ^†^ Subjects without any related reports were excluded from the analysis. * *p* values are based on the Wilcoxon signed rank test. *P1* = comparison between MFS and FS; *P2* = comparison between MFS and washout; *P3* = comparison between FS and washout; significant differences at *p* < 0.05.

## Data Availability

The study data are available from the corresponding authors upon request.

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
