# Peer review of "Tolerability of Oral Supplementation with Microencapsulated Ferric Saccharate Compared to Ferrous Sulphate in Healthy Premenopausal Woman: A Crossover, Randomized, Double-Blind Clinical Trial"

_ijms, 2022, doi:10.3390/ijms232012282_

Round 1
Reviewer 1 Report
This is a well-written description of a well-designed tolerabilty study of a commercial formulation for iron supplementation. My critique is confined to the choice of exclusively Caucasian subjects of normal BMI, hardly a typical patient population.
Reviewer 2 Report
The manuscript "Tolerability of oral supplementation with microencapsulated ferric saccharate compared to ferrous sulphate in healthy premenopausal woman: a crossover, randomized, double-blind clinical trial" lacks a molecular objective to be published in the International Journal of Molecular Science. The goal of the study is a clinical trial and molecules are not the object of study. The authors used a commercial medicine that at least was previously tested clinically to put a new formulation in the pharmaceutical market. The authors did not have tested a novelty idea to microencapsulate ferric to treat or prevent anemia.
In the Scope of the International Journal of Molecular Sciences, molecules are the object of study; among those studies, we find:
- fundamental theoretical problems of broad interest in biology, chemistry and medicine;
- breakthrough experimental technical progress of broad interest in biology, chemistry and medicine; and
- application of the theories and novel technologies to specific experimental studies and calculations.
For that I cannot recommend the publication in IJMS.
Reviewer 3 Report
In this manuscript Friling et al., based on a single center, crossover, randomized, double-blind and controlled clinical study, demonstrate a better tolerability profile of a microencapsulated ferric saccharate formulation over ferrous sulphate.
The study is clear and the results are properly reported. However, there is just a bit of space for improved clarity:
- What was the reason for the choice of a 2-way crossover trial? What were the authors trying to capture in terms of limitations/outcomes?
- Why was ferrous sulphate and not a matched substance correspondent (iron saccharate) used as a control?
- I think authors should be careful when reporting the meaningfulness of a result. This was particularly clear in 2.3.3 when an increased overall daily intensity of heartburn for MFS compared to the washout period was deemed not clinically meaningful due to the small sample size and low intensity level. Given that the sample size is transversal to the whole study it is not clear why was this specific result not meaningful.
- I think that the manuscript would benefit from a small paragraph in the end of the discussion referring study limitations.
Reviewer 4 Report
The manuscript presents the results of observations of the effect of iron administration reported by volunteers participating in the study. The tested microencapsulated iron is a product available on the market, in the description of which the manufacturer states that it has tested the effects of uptake and no negative observations :“… AB-Fortis® uses and activity are supported by published scientific research. A human clinical trial showed the high bioavailability of the iron in AB-Fortis® in comparison with iron sulfate1. Increased bioavailability of the ingredient may lead to higher blood levels of iron achieved with daily intake. Reduced interactions of AB-Fortis® within the gastrointestinal system reduces iron-related adverse effects such as metallic taste2. [https://iff-health.com/portfolio/ab-fortis-pp/ ]“
Is the testing intention to verify this manufacturer's declaration and previous research results?
The manuscript does not meet the journal aims and scope. It does not present a research suitable for an advanced forum for molecular studies in biology and chemistry, with a strong emphasis on molecular biology and molecular medicine.
Minor comments:
As I understand it, the column 2 represents the mean of the data from all subjects, whereas columns 3 and 4 represent the group averages. If so, how is it possible that the mean of the total of respondents differs so much from the mean value of the groups?
table 1. :
|
Ferritin level > 30 μg/L, n (%) |
// 19.0 (40.0) |
// 9.0 (37.5) |
// 10 (43.5) |
Data expressed as mean (standard deviation) - standard deviation higher than the mean value?
Round 2
Reviewer 2 Report
I agree in part with the response of the authors. I think the molecular approach could be improved in the manuscript. Despite my opinion about molecular research, the manuscript can be contributed to the development of a new formulation of iron to avoid gastrointestinal disturbance in women's iron deficiency.
Reviewer 4 Report
I stand by my opinion, that the manuscript does not meet the journal aims and scope. The editorial corrections included in the 2nd version do not change the character of the work.
Additionally, I do not support the idea of duplication the work carried out (by one of the authors) about 10 years ago.
